# Study on the Aqueous CdTe Quantum Dots Solar Device Deposited by Blade Coating on Magnesium Zinc Oxide Window Layer

**DOI:** 10.3390/nano12091523

**Published:** 2022-04-30

**Authors:** Bin Lv, Xia Liu, Bo Yan, Juan Deng, Fan Gao, Naibo Chen, Xiaoshan Wu

**Affiliations:** 1Collaborative Innovation Center for Bio-Med Physics Information Technology of ZJUT, Zhejiang University of Technology, Hangzhou 310023, China; 2111909018@zjut.edu.cn (X.L.); jdeng@zjut.edu.cn (J.D.); gaofan@zjut.edu.cn (F.G.); chennb@zjut.edu.cn (N.C.); 2Nantong-Nanjing University Institute of Materials Engineering & Technology, Nantong 226019, China; xswu@nju.edu.cn; 3Department of Applied Physics, Zhejiang University of Technology, Hangzhou 310023, China; 4National Laboratory of Solid-State Microstructures, Nanjing University, Nanjing 210093, China

**Keywords:** aqueous CdTe quantum dots, solar cell, blade coating, Zn_1−x_Mg_x_O

## Abstract

Aqueous CdTe quantum dots solar cells have been successfully fabricated by the blade coating method on the magnesium zinc oxide (Zn_1−x_Mg_x_O or ZMO) window layer. Compared with the ZMO mono-window layer, the ZMO/CdS bi-window layer can decrease the interface recombination effectively due to the lower lattice mismatch and fast interdiffusion between CdS and CdTe. Moreover, the high temperature annealing of the CdTe quantum dots absorbed layer passivates the grain boundary of the CdTe crystalline via the replacement reaction of tellurium with sulfur. Finally, the conversion efficiency of our aqueous CdTe quantum dots solar device is improved from 3.21% to 8.06% with the introduction of the CdS interlayer and high temperature CdCl_2_ annealing. Our aqueous CdTe quantum dots solar devices show a large open circuit voltage and fill factor which are comparable with the conventional devices that are fabricated with organic CdTe quantum dots. We believe that it is the spike-like conduction band alignment between the ZMO and CdTe absorbed layer that reduces the majority carrier concentration, leading to the decrease in interface recombination probability.

## 1. Introduction

CdTe has been considered the potential market-leading photovoltaic material with a large absorption coefficient. Many technologies have been developed for fabricating CdTe solar cells with laboratory efficiencies higher than 20% [1], such as close-spaced sublimation [2]. However, most vacuum deposition methods require high temperature and gas protection, which is expensive and complex. Due to numerous advantages, including controllable stoichiometry, melting point depression, and low-cost procedure, the solution processing technology based on colloidal CdTe quantum dots (QDs) [3] may offer a way to solve this problem.

However, the quantum dots prepared by the organic synthesis method confront problems such as high toxicity of organic ligands and unfriendliness to the environment. On the contrary, the aqueous synthesis method does not have these problems. Water is undoubtedly the most popular method because of its low cost and environmentally-friendly characteristics. To date, the performance of aqueous synthesized CdTe QDs solar cells has been largely improved through various routines [4,5,6]. However, the conversion efficiency of the CdTe QDs solar device is still far behind traditional polycrystalline CdTe solar cells.

As is well known, the microstructure of a CdTe film is very important due to its optical and electrical properties, which affect the device’s performance. As to the vacuum deposition method, the CdTe film usually consists of micro-sized grains with relatively few grain boundaries and a smoother morphology, which is beneficial for photoelectric conversion [7]. In contrast, blade coating [8] and spin-coating [9] are commonly used in CdTe QDs solar cells’ fabrication. Recently, several research groups developed the blade coating method to improve the performance of electrical devices based on QDs [10,11,12,13]. However, the device based on QDs is always full of grain boundary defects due to the nanoscale size of QDs. The nanostructure will increase interface and bulk recombination in the heterojunction, which strongly suppresses the conversion efficiency of PV devices. Therefore, learning how to improve the quality of the aqueous CdTe QDs absorbed layer deposited with blade coating is crucial to improving the performance of solar devices.

Recently, magnesium zinc oxide (ZMO) has been used as a window layer in high efficiency CdTe polycrystalline thin film solar cells. The ZMO/CdTe solar cell with the maximum solar conversion (22.1%) [1] efficiency was made by First Solar in 2016 [14]. Compared with the conventional CdS window layer, ZMO has a much larger optical band gap, possibly resulting in more photons in the short wavelength range passing through and being absorbed by the CdTe active layer. On the other hand, compared with the large band gap window layers, such as ZnO and SnO_2_, the ZMO can achieve favorable band alignment, which is called “spike”, with CdTe to maximize the built-in potential and minimize the interfacial recombination at the window/absorber interface [15].

Inspired by the previous report about the crystalline CdTe solar cells [6], we introduce a ZMO window layer instead of ZnO [16], which has been frequently used in CdTe QDs solar devices. Furthermore, we add the CdS interlayer between ZMO and CdTe, which decreases the interface recombination further. Meanwhile, the CdCl_2_ annealing treatment is also introduced into the fabrication process of aqueous CdTe QDs solar cells. The high temperature annealing process plays an important role in decreasing bulk recombination. Our aqueous CdTe QDs solar devices show a large open circuit voltage and fill factor, which are comparable to the conventional devices that were fabricated with organic CdTe QDs [3]. We attribute this to the use of the ZMO and CdS polycrystal window layer, which can easily establish the spike-like conduction band alignment at the interface of heterojunction.

## 2. Materials and Methods

Our solar devices were fabricated on the commercial glass coated with Indium Tin Oxide (ITO). The basic structure of our CdTe QDs solar devices was Glass/ITO/ZMOZMO/CdTe(QDs)/Au. Moreover, a CdS interlayer was deposited between ZMO and CdTe QDs results in the structure of Glass/ITO/ZMOZMO/CdS/CdTe(QDs)/Au to study its effect on the performance of solar cells. 

Step 1: The commercial glass coated with 150 nm of ITO was used as substrate. A 120 nm ZMO layer was deposited by RF magnetron sputtering using a JPG500 sputtering system with the base pressure of 5 × 10^−4^ Pa. The depositions were performed from Zn_1−x_Mg_x_O target with a composition of 9 wt% Mg with 91% wt% Zn under mixed flow gas of 3% oxygen and 97% argon. The pressure during the sputtering process was kept at 1 Pa and the applied sputtering power was 200 W.

Step 2: Only for the solar cell with the structure of Glass/ITO/ZMO/CdS/CdTe(QDs)/Au, the nano-size (~15nm) CdS films were grown by chemical bath deposition with hydrogen peroxide (H_2_O_2_) in the solution as we previously reported [17]. The thickness of CdS layer was controlled at 20nm. In order to enhance the n-type of ZMO and CdS, the samples of ZMO and ZMO/CdS were then annealed at 400 °C for 30 min with vacuum ambiance (10^−1^ Pa).

Step 3: The thiol-capped CdTe QDs were synthesized with the mixture of solution NaHTe, CdCl_2_, NaOH, and Thioglycolic acid (TGA) in three-necked flask stirring to reflux (90 °C) under atmospheric conditions. The details of QDs synthesis can be found in our previous work [18]. We finally choose the QDs that refluxed after 16 h to deposit the absorbed layer for solar cells.

Step 4: The CdTe QDs absorber layer was blade coated on ZMO mono-window layer or ZMO/CdS bi-window layer depending on the device structure. Before the blade coating process, the chosen CdTe QDs’ dispersion was taken to centrifuge for 20 min with acetone. The obtained precipitations were gathered and dropped on substrate and blade coated evenly using a 10 μM wire rod at the speed of 100 mm/min. Then, the substrate was annealed at 150 °C for 8 min and 300 °C for 30 s to remove the organic solvent and impurities. After repeating the above process 10 times, the approximate 1.2 μm CdTe QDs absorber layer accompanied with substrate was then annealed at 150 °C for 3 min, followed by dipping into saturated CdCl_2_ methanol solution and rinsing in acetone for 10 s to remove excess CdCl_2_. The multilayer was then annealed at 250 °C or 350 °C for 2min, respectively, and cooled down to room temperature. The multilayers annealed at low temperature were labeled as ZMO250 and ZMO/CdS250. Meanwhile, the multilayers annealed at high temperature were labeled as ZMO350 and ZMO/CdS350.

Step 5: All multilayers were etched in 0.2% Br/CH_3_OH solution for 8 s to remove surface oxides and to create a Te-rich surface according to the chemical reaction: Br_2_ + Te^2−^ = 2Br^1−^+Te [19]. After etching, a 100 nm Au contact was deposited by magnetron sputtering. After annealing in Ar at 250 °C, the solar cells were completed with conductive silver painting.

An atomic force microscope (AFM, Veeco Dimension 3100, Santa Barbara, CA, USA) with the tapping mode was used to observe the morphology of the ZMO and ZMO/CdS films. The thicknesses of ZMO and CdS films were measured by VeecoDektak 6M stylus surface profilometry (Santa Barbara, CA, USA). Optical properties of window layer and CdTe QDs absorbed layer were detected in the range of 300–1000 nm using UV–vis spectrophotometer (Varian, Cary50Probe, Palo Alto, CA, USA). The structure of the films was characterized by X-ray Diffraction (XRD, Siemens, D5000H, Munich, Germany). The morphological surface of CdTe was examined by scanning electron microscope (SEM, 1530VP, Oberkochen, Germany). The microstructures and energy-dispersive x-ray (EDX) spectra of CdTe QDs films were investigated using high resolution transmission electron microscopy (HRTEM, FEI Tecnai G2 F30, Hillsboro, OR, USA). The external quantum efficiency (EQE) spectrum of solar cell was obtained by quantum efficiency measurement system (PV Measurements QEX10, Point Roberts, WA, USA) and the conversion efficiency was measured by I–V test system (Newport PVIV, Irvine, CA, USA).

## 3. Results and Discussion

### 3.1. Structural and Optical Properties of Window Layers

Figure 1a shows XRD patterns for ZMO and ZMO/CdS films after vacuum treatment. The identification and assignment of the observed diffraction patterns are made using the data from the Joint Committee for Powder Diffraction Standard (JCPDS). As are shown both in ZMO and ZMO/CdS films, the (0 0 2) and (0 0 4) reflections at 32.80° and 72.80° belong to the hexagonal phase of ZMO [20]. Meanwhile, the ZMO/CdS bilayer film shows the characteristic peaks (002) of CdS with hexagonal phase at 26.6°. The obvious signal of the CdS diffraction indicates good crystallization because only 20 nm CdS is deposited on ZMO (Appendix A). The good crystallization is expected because the hexagonal phase is thermodynamically stable for CdS [17,21], which leads to a preferred orientation at (0 0 2).

The AFM images are shown in the inset of Figure 1a. The roughness of the ZMO monolayer and the ZMO/CdS bilayer are 15.8 and 12.6 nm, respectively. Cracks among grain boundaries can be found in the ZMO 3D image. These are evidently the reasons for reducing the device open circuit voltage and fill factor. On the contrary, the ZMO/CdS bilayer film has a nanosized CdS (~20 nm) on top, leading to fewer pinholes and cracks among grain boundaries. The surface is smoother than that of the ZMO single layer due to the gap filling between ZMO grains by nanosized CdS. As a window layer, ZMO/CdS, with fewer holes and a more uniform grain size distribution, has a positive effect on the performance of the final device.

The optical transmittance spectra of ZMO and ZMO/CdS films after vacuum treatment are shown in Figure 1b. We note that the transmittance of the ZMO single layer is much larger than that of ZMO/CdS bilayer in the short wavelength range before 520 nm. This means that more photons in the short wavelength range may pass through the window layer and contribute to the photocurrent in the solar cells. The absorption edge of CdS is inconspicuous due to the interference effect of the ZMO layer. The optical pictures of ZMO and ZMO/CdS films are shown in the inset of Figure 1b. Compared with the ZMO film, the ZMO/CdS film shows a light-yellow color, which corresponds to the optical absorption above the band gap of CdS. This may explain why the ZMO film has a larger transmittance in the short wavelength.

### 3.2. Quality of CdTe QDs

To fabricate the solar devices based on aqueous CdTe QDs, with as large as possible photocurrent, we chose the QDs with the largest size among our samples prepared with different reflux time than the materials of the absorbed layer. The evolution of both absorption and photoluminescence (PL) spectra of CdTe NPs as a function of reflux time are presented in Appendix A. As the reflux proceeds, the PL peak position can approach 546, 562, 598, and 618 nm in the period of growth time of 5, 8, 12, and 16 h, respectively. Here, the QDs exhibit a PL peak at 618 nm; they were chosen as the absorbed materials in the following devices. As shown in Figure 2a, the absorbance spectra of QDs have a sharp first excitonic absorption peak at 567 nm, corresponding to the quantum confinement effect (QCE). The characteristic size of CdTe QDs is calculated to be 3.84 nm by the empirical functions proposed by Peng et al. [22], with the first absorption peak positions in absorption spectra:(1)$$D=(9.8127\times 10^{-7})\lambda^{3}-(1.7147\times 10^{-3})\lambda^{2}+(1.0064)\lambda -(194.84).$$

The calculated size (3.84 nm) is consistent with the value (3.87 nm) estimated from the HRTEM image shown in the inset of Figure 2b. Moreover, the small Stokes shift (about 170 meV) between the excitonic absorption edge and the PL emission peak indicates the good quality of our QDs.

### 3.3. Optical Property of CdTe QDs Layer

Based on the ZMO and ZMO/CdS window layer, we deposit CdTe QDs by the blade coating method to form optoelectronic heterojunctions. Then, we complete the solar devices with the structure of Glass/ITO/ZMO/CdTe(QDs)/Au and Glass/ITO/ZMO/CdS/CdTe(QDs)/Au. The entire fabrication process and device configuration of solar cell are shown in Figure 3a with a cross-sectional SEM image of the actual multilayer with the structure Glass/ITO/ZMO/CdS/CdTe(QDs). The thickness of each layer is marked in the image. As is well known, the annealing process is crucial to improving the CdTe device performance [23], especially for solution-processed CdTe QDs solar cells, which was confirmed by Qin et al. [9]. Thus, both multilayers with two different structures were deposited and CdCl_2_ annealed at low (250 °C) and high (350 °C) temperature to study the effect of the annealing process on the performance of the corresponding devices. Figure 3b shows the absorbance spectra of these four multilayers. ZMO250 and ZMO350 represent the multilayers with the same structure of Glass/ITO/ZMO/CdTe(QDs) but different annealing temperature (250 °C and 350 °C). Meanwhile, ZMO/CdS250 and ZMO/CdS350 represent the multilayers with the same structure of Glass/ITO/ZMO/CdS/CdTe(QDs) but different annealing temperature. The absorbance (*A*) is calculated with transmittance (*T*), using the formula:(2)A=log(1/T).

Moreover, because the thickness (Figure 3a) and the absorption coefficient of the CdTe QDs layer are much larger than that of ITO, ZMO, and CdS [20,23], the absorbance signal of the multilayer is dominated by the CdTe QDs layer. Thus, the optical band gap *E_g_* of the CdTe QDs layer can be determined by Tauc’s plots of the multilayer by the intersection of the tangent to the *hv* axis using the below expression:(3)$${\left(Ahv\right)}^{2}\propto (hv-E_{g}),$$
where *d* is the thickness, *T* is the transmittance of the multilayers, *E_g_* is the optical band gap of CdTe QDs, *A* is absorbance, and *hv* is the photon energy. As shown in the inset of Figure 3b, firstly, the optical band gap of the CdTe QDs layers (1.54~1.57 eV) is close to the value of their bulk materials [24] (1.48 eV) which is much smaller than that of the corresponding CdTe QDs (2.17 eV). Furthermore, there is no obvious excitonic peak near the absorption edge, which implies the decline of QCE. Secondly, the absorption edge of the CdTe QDs layer does not significantly change with the CdS interlayer between ZMO and CdTe QDs, indicating that the CdS interlayer does not affect the optical properties of the CdTe QDs absorbed layer. Finally, the high temperature annealing CdTe QDs layers have a lower band gap (1.57 eV) than the low temperature annealing ones (1.54~1.55 eV), which can greatly help to improve the device photocurrent theoretically. We attribute this to the “bowing effect” of the band gap [25] due to the formation of the CdTe_1−x_S_x_ alloy [26] in the CdTe QDs layer during the annealing process, leading to the red shift of optical absorption. The lattice constant obtained from HRTEM images and the elemental analysis by the EDX test of the CdTe QDs layers prove our theory, which we will discuss below.

### 3.4. Structural and Elemental Analysis of CdTe QDs Layer

Figure 4 shows the SEM micrographs of the CdTe QDs layer of four multilayers that were mentioned above. According to the top view image, the crystalline grain size of these CdTe QDs films is estimated to be about 20~30 nm with the naked eye. Furthermore, compared to the 250 °C-annealing films (ZMO250 and ZMO/CdS250), the 350 °C-annealing films (ZMO350 and ZMO/CdS350) have a much more indistinct grain boundary. This means the boundary barriers are smoothed more effectively by high temperature annealing.

The inset of Figure 4 shows XRD patterns of the CdTe QDs layer in these four multilayers. All CdTe QDs layers are textured polycrystalline with the cubic phase. The (111) and (511) texture of the films increased obviously in the high temperature annealing. The texture coefficient Ci can be determined as [7]
(4)$$C_{i}=\frac{I_{i}/I_{0i}}{(1/N)\sum_{i}^{N}\left(I_{i}/I_{0i}\right)},$$
where, *I_i_* is the intensity of a generic peak in the pattern, *I_0i_* is the intensity of a generic peak for a standard powder sample (JCPDS), and *N* is the number of reflections calculated in the XRD pattern. Seven main diffraction reflections are taken into account in the present. Then, the preferred orientation factor *f* of a film is calculated from the formula
(5)$$f=\sqrt{\frac{\sum_{i}^{N}{\left(C_{i}-1\right)}^{2}}{N}},$$

If *f* equals 0, the grain orientation of the sample is the same as that of the standard powders, which indicates that the polycrystalline film has almost perfect standard powders. From the calculation results of *f* shown in the XRD pattern of Figure 4, we conclude that the high temperature annealing may sharply increase the preferential orientation, which will contribute to the reduction in carrier recombination in future devices [27]. As is well known, the addition of CdCl_2_ in the annealing process will induce re-crystallization, grain growth, and surface smoothening [2,28]. During annealing, the recrystallization and orientation reconstitution occurred.

Besides recrystallization, for high temperature annealing films, the characteristic diffraction peaks exhibit a small amount of shifting compared with the corresponding peaks of low temperature annealing films. As can be seen from the XRD patterns, the (1 1 1) peaks of the CdTe QDs layer in ZMO250 and ZMO/CdS250 are located at the same angle, 24.06°. Meanwhile, the corresponding peaks in the ZMO350 and ZMO/CdS samples shift to 24.36° and 24.42°, respectively. We attribute this shifting to the formation of the CdTe-like alloy, CdTe_1−x_S_x_, which reduces the lattice constant of crystal, which has also been suggested in the discussion of multilayers’ optical absorption.

Finally, the individual crystalline size *t* of CdTe QDs films was determined using Scherrer’s formula
(6)t=(Kλ)/(Bcosθ),
where *K* is Scherrer’s constant, which is a reference value corresponding to the quality factor of the apparatus measured with a reference single crystal and dependent on the crystallite shape (0.89–0.9). λ is the X-ray wavelength, *B* is the full width at half maximum (FWHM) of the diffraction peak, and θ is the Bragg angle. The FWHMs of (1 1 1) peaks and crystalline size t are put in order in Table 1.

According to the result of Table 1, the crystalline size t of the CdTe QDs film of these four multilayers is around 10~12 nm, which is approximately half of the value of the grain size estimated with the naked eye from the top view of SEM images. This indicates that the grain is not a single crystalline. Furthermore, this may explain the polycrystalline-like texture of CdTe QDs films. It worth noting that, compared with the low temperature annealing sample, the crystalline size decreases with high temperature annealing, which indicates the decrease in crystallinity slightly. It is believed that CdCl_2_ annealing can generate the grain growth of the CdTe QDs film [16]. On the contrary, we believe that the sulfur loss during high temperature annealing combines with the recrystallization results in the decrease in crystalline size.

In order to study the above hypothesis, we carried out HRTEM and EDX measurement for the CdTe QDs layer. Figure 5a,b show the HRTEM images of the CdTe QDs layers deposited on the ZMO/CdS window layer suffering low and high temperature annealing. As shown in the figure, the (1 1 1) interplant distance of CdTe crystalline was measured and denoted. The (1 1 1) interplant distance of CdTe (~0.369 nm) suffering low temperature annealing is close to the value of CdTe QDs (~0.368 nm) measured in QDs dispersion, as shown in Figure 2b. In contrast, the (1 1 1) interplant distance of CdTe (~0.363 nm) suffering high temperature annealing decreases obviously. Considering the method we used to prepare CdTe QDs, we attribute the decrease in the lattice constant of the CdTe to the replacement reaction of tellurium with sulfur, which is released from TGA ligands at the surface of QDs during the high temperature annealing process [18]. We determined this in our previous study.

On the other hand, the TGA ligands tend to gather around the CdTe crystalline due to the oriented attachment growth of CdTe grains in the period of blade coating deposition. The resolve of TGA during high temperature annealing results in the sulfur escaping from the crystalline surface beside the replacement reaction, which decreases the grain size of CdTe; this is presented in the SEM image and the diffraction peak shift in Figure 4. The EDX spectra in Figure 5c,d also supports the above mechanism. The results of the stoichiometry detected by the EDX measurements clearly illustrate the decrease in the ratio of sulfur after high temperature annealing. It is worth noting that the ratio of sulfur in CdTe QDs film deposited on the ZMO window layer is obviously less than that deposited on the ZMO/CdS window layer. It might be the fast diffusion of sulfur from the CdS interlayer at the annealing process that increases the ratio of sulfur in CdTe QDs film deposited on the ZMO/CdS window layer.

### 3.5. Photovoltaic Measurements

The effect of the ZMO mono-window layer and the ZMO/CdS bi-window layer on the carrier collection is investigated with the EQE spectrum. As shown in Figure 6a, the EQE of the device based on the ZMO/CdS bi-window layer (ZMO/CdS250 and ZMO/CdS350) are larger than that based on the ZMO mono-window layer (ZMO250 and ZMO350) at short wavelength before 650 nm. This is an unexpected result, because the transmittance of the mono-window layer is larger than that of the bi-window layer, especially in the short wavelength range (Figure 1b). In order to isolate the influence of the carrier collection on the EQE, we calculate the specific value of transmittance of the ZMO mono-window layer to the ZMO/CdS bi-window layer (*T*_ZMO_/*T*_ZMO/CdS_) and the specific value of EQE based on these two different window layers (EQE_ZMO_/EQE_ZMO/CdS_) in the range of the entire wavelength. As shown in Figure 6b, the transmittance ratio, *T*_ZMO_/*T*_ZMO/CdS_, is quite large before 520 nm due to the optical absorption above the band gap of CdS, as expected. However, both the EQE ratio of devices annealing at low temperature (EQE_ZMO250_/EQE_ZMO/CdS250_) and high temperature (EQE_ZMO350_/EQE_ZMO/CdS350_) exhibit a value that is smaller than 1. This means the photoelectric conversion of the device based on the ZMO mono-window layer is less than that based on the ZMO/CdS bi-window layer.

A more detailed discussion requires the analysis of the carrier collection over the absorbed layer. EQE is affected by the carrier collection as the following formula [23]:(7)$$EQE(\lambda )=\sum_{z}T(\lambda )\exp[-\alpha (\lambda )z]g(z),$$
where *T*(*λ*) is the transmittance of the window layer, *α*(*λ*) is the absorption coefficient, and *z* denotes the position where the photons arrive. *g*(*z*) is the carrier collection probability, which is changed with z due to the interface and bulk recombination. Generally speaking, the absorption coefficient *α*(*λ*) always decreases with increasing wavelength. This means that most ultraviolet light can be absorbed and stimulate the photoelectrons near the interface between the window layer and the absorbed layer. On the contrary, the light with long wavelength can arrive at the depth of the absorbed layer and stimulate the photoelectrons. Thus, EQE at a short wavelength range always reflects the carrier collection properties near the interface of heterojunction. Meanwhile, EQE at long wavelength range reflects the carrier collection properties in the absorbed layer. Moreover, it is well known that the interface recombination dominates carrier collection at the interface between the window layer and the absorbed layer; meanwhile, the bulk recombination dominates it at a deep range of the absorbed layer [29].

As can be seen from Figure 6b, both the EQE ratio of devices annealing at low temperature (EQE_ZMO250_/EQE_ZMO/CdS250_) and high temperature (EQE_ZMO350_/EQE_ZMO/CdS350_) increases with increasing wavelength during 300–400 nm. Considering the similar absorbance (Figure 3b) of the CdTe QDs layer in all four of these devices, we conclude that the ZMO/CdS bi-window layer can reduce the interface recombination effectively compared with the ZMO mono-window layer. This conclusion can also be proved by the EQE spectra in Figure 6a, where the ZMO/CdS250 device has a larger EQE than the ZMO350 device even when the latter suffers the higher temperature annealing.

Furthermore, the EQE ratio approaches one at long wavelength range 650–800 nm for all devices. This indicates that the window layer has little effect on the bulk recombination in the CdTe QDs absorbed layer. In contrast, high temperature annealing effectively improves the carrier collection in the bulk range of the absorbed layer by reducing the bulk recombination, which is proved by the EQE spectra at long wavelength range in Figure 6a. As is discussed in the morphology and structural properties of the CdTe QDs layer, in the period of annealing, the thiol ligands will be resolved and release sulfur to replace the tellurium in the CdTe QDs generating CdTe_1−x_S_x_ alloy in the QDs. The sulfur can effectively passivate the dangling bond of CdTe QDs [30,31], showing a similar effect as the CdS shell [32]. Moreover, a red shift in the absorption edge near 800 nm in the EQE of the high temperature annealed device also proves the formation of the low band-gap alloy CdTe_1−x_S_x_, which is consistent with the absorbance measurement in Figure 3b. Finally, the photocurrent density *J*_L_, calculated from the EQE spectra and combined with the reference solar spectral irradiance at the ground (AM1.5) spectra, for all devices is listed in Figure 6a.

The effect of the ZMO/CdS bi-window layer on the CdTe QDs solar cell is also studied with the conversion efficiency test, as shown in Figure 7a. The detailed parameters of the devices are listed in Table 2. The short circuit current density *J*_s_ of these four devices takes the similar value of photocurrent density *J*_L_ in the EQE measurement for corresponding devices. However, the devices based on the bi-window layer have a larger open circuit voltage *V_oc_* and fill factor *ff*, compared with the devices based on the mono-window layer. This further proves our conclusion that the bi-window layer can decline the interface recombination, which improves the minority carrier lifetime. Moreover, the high temperature annealing of the CdTe QDs layer also improves the *V_oc_* and *ff* compared with the low temperature annealing devices. We attribute this to the passivation effect of sulfur to the dangling bond of CdTe QDs, which also improves the minor carrier lifetime.

It is worth noting that the annealing treatment does not change the series resistance *R*_s_ and shunt resistance *R*_sh_ obviously. Furthermore, the value of series resistance *R*_s_ of the QDs solar device is much larger than that of our polycrystal film solar devices [33]. The reason might be the QDs have more grain boundary and a weaker link between single crystalline compared with the polycrystal film. Meanwhile, the ZMO/CdS bi-window layer reduces the current leakage remarkably, which is manifested by the distinct increase in shunt resistance *R*_sh_ from less than 300 to more than 1000 Ωcm^2^. We attribute this to the lower lattice mismatch between CdS and CdTe than that between ZMO and CdTe. The lower mismatch results in a more compact heterojunction, which has fewer leakage channels. This is another factor that improves the *ff* and *V_oc_*.

In order to distinguish the recombination pathway of the CdTe QDs solar device based on different window layers, the *V_oc_* was measured as a function of temperature in the range of 288–348 K, which is shown in Figure 7b. The ideal expression of the open-circuit voltage is
(8)$$V_{oc}=\frac{E_{a}}{q}-\frac{nkT}{q}\ln(\frac{J_{00}}{J_{0}}),$$
where *J*_0_ is the diode current density, *n* is the diode ideality factor, *k* is the Boltzmann constant, *T* is the temperature, *E_a_* is an activation energy associated with recombination in solar cells, and *J*_00_ is the reference current density which is temperature-independent. Depending on the recombination pathways, different values for *n*, *J*_00_, and *E_a_* can be obtained. The activation energy *E_a_* is important to distinguish the dominant recombination path between interface and bulk. For a heterojunction of high quality, the activation energy is close to the bandgap of the CdTe QDs absorber, that is, the recombination is the bulk mechanism. However, lower activation energy can also result in recombination at the interface due to conduction band misalignment [34].

As shown in Figure 7b, the *V_oc_* versus temperature follows a straight line well for all devices and the mean measurement error (≈0.3%) was calculated by Root-Mean-Square (RMS) with respect to the deviation between the straight line and the data for these four devices. All of the extrapolation results of activation energy are lower than the bandgap of the CdTe QDs layer (1.54–1.57 eV obtained from Figure 2b). The obtained value of *E_a_* for the devices with the ZMO mono-window layer is much smaller than the bandgap, only about 1.01 and 1.12 eV for the low and high temperature annealing device, respectively. This indicates that the dominant recombination is interface recombination. The value of *E_a_* for the devices with the ZMO/CdS bi-window layer is 1.38 and 1.42 eV for low and high temperature annealing device, respectively. The value is close to the bandgap of the CdTe QDs layer, indicating that the dominant recombination is bulk recombination. Compared with devices with ZMO and ZMO/CdS window layers, the dominant recombination has changed from interface recombination to bulk recombination and the *V_oc_* of the devices has been improved significantly. As shown in Figure 7c, compared with the conventional CdS [9], the ZMO window layer can generate the spike-like conduction band alignment at the heterojunction interface, where the conduction band of the window layer is higher than that of the CdTe layer [15]. The spike-like band alignment reduces the diode current density *J*_0_ by lessening the electron concentration due to the higher conduction band in the ZMO side. Meanwhile, this band alignment drives the electrons to the CdTe side and recombines them with the hole through bulk mechanism (blue route in Figure 7c). However, the ZMO mono-window layer does not suppress the recombination obviously because of the defect energy state (*E*_D_), which is produced by the relatively large lattice mismatch between ZMO and CdTe [15]. The defect states at the interface open an interface recombination route in parallel with the bulk recombination route, which reduces the activation energy *E_a_* to approximately 1.0 eV. In contrast, the ZMO/CdS bi-window layer diminishes the defect states due to the lower lattice mismatch and the interdiffusion between CdS and CdTe, thus suppressing the interface recombination and increasing *E_a_* to approximately 1.5 eV.

## 4. Conclusions

In conclusion, the CdTe solar cells based on the aqueous CdTe QDs absorbed layer have been successfully fabricated by the blade coating method on the ZMO window layer. We found that the CdS interlayer between ZMO and CdTe plays an important role in the interface recombination. The ZMO/CdS bi-window layer can reduce the interface recombination effectively due to the lower lattice mismatch and fast interdiffusion between CdS and CdTe, which passivates the dangling bond of CdTe QDs. In contrast, a high temperature annealing process play an important role in the bulk recombination. The thiol ligand TGA used in aqueous phase synthesis of CdTe QDs provides lots of sulfur to passivate the grain boundary of the CdTe crystalline via the replacement reaction of tellurium with sulfur during the high temperature annealing process. Our aqueous CdTe QDs solar devices show a large open circuit voltage and fill factor, which are comparable with the conventional devices that were fabricated with organic CdTe QDs. We attribute this to the use of the ZMO and CdS polycrystal window layer, which can easily establish the spike-like conduction band alignment at the interface of heterojunction. The spike-like alignment can reduce the majority carrier concentration that decreases the recombination probability between the majority carrier and minority carrier. Generally, the conversion efficiency of our CdTe QDs devices is not as good as the polycrystal CdTe film solar devices, mainly due to the inefficiency of photoelectron collection. The exploration in this work provides valuable solutions for the low-cost and nontoxic fabricating process of aqueous QDs solar cells.

## Figures and Tables

**Figure 1 nanomaterials-12-01523-f001:**
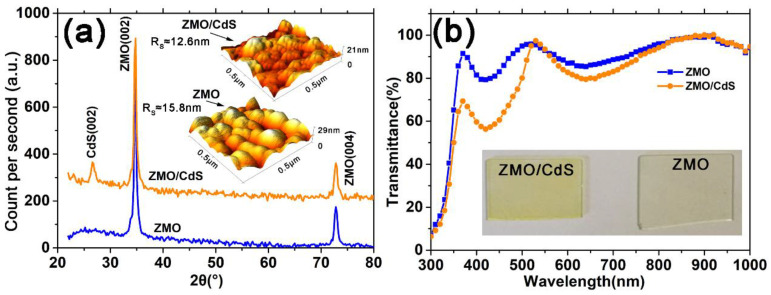
(**a**) XRD pattern for ZMO and ZMO/CdS films during 20°–80°. All ZMO and CdS have the hexagonal phase. The inset is the AFM images of corresponding films. (**b**) Transmittance spectra of ZMO and ZMO/CdS films. The inset is the optical pictures of corresponding films.

**Figure 2 nanomaterials-12-01523-f002:**
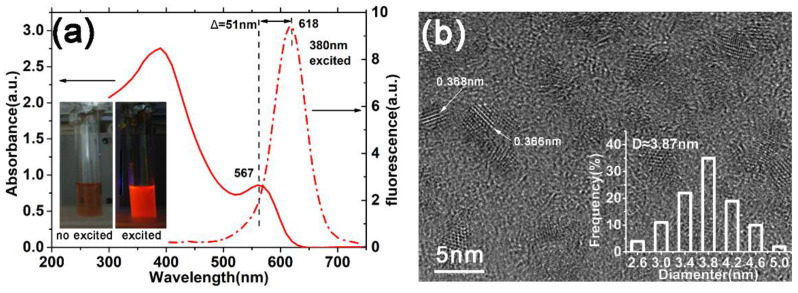
(**a**) The absorption and photoluminescence spectra of the CdTe QDs, which are used in our solar device. The inset shows the photograph of the NPs with and without UV irradiation. (**b**) The HRTEM images of corresponding QDs in (**a**). The size distribution of QDs is marked in this figure.

**Figure 3 nanomaterials-12-01523-f003:**
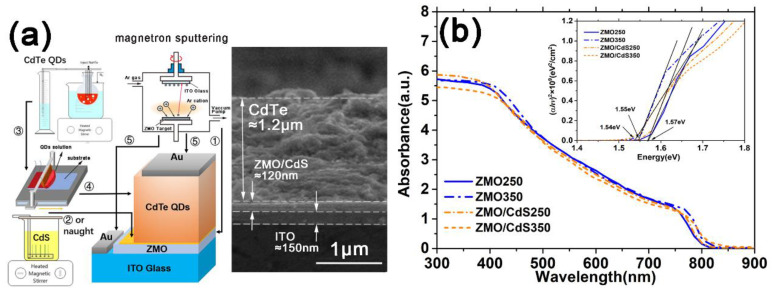
(**a**) The fabrication schematic and device configuration of the CdTe QDs solar cell. Each step is labeled with sequence number. The cross-sectional SEM image of a typical multilayer device without Au electrode is shown beside the schematic. (**b**) The absorption spectra of multilayer based on CdTe QDs absorbed layer without Au electrode suffering different annealing temperature. The inset shows the optical band gap obtained from Tauc’s plots of corresponding samples.

**Figure 4 nanomaterials-12-01523-f004:**
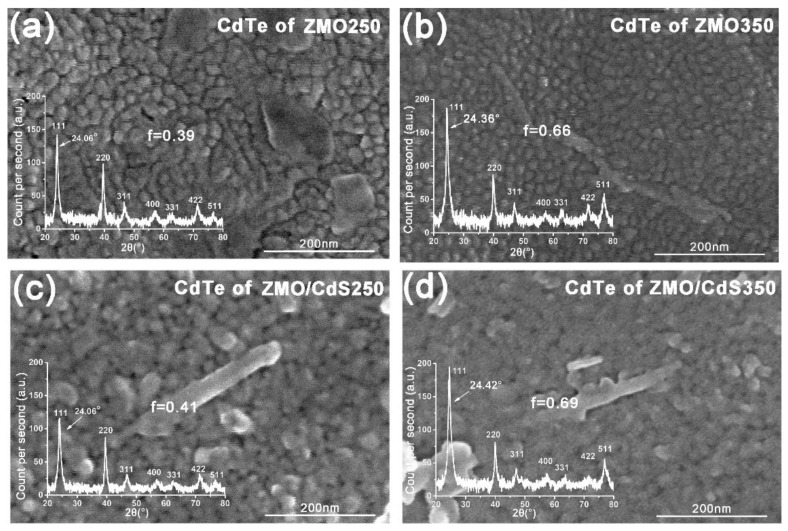
The top view SEM images of CdTe QDs layer in multilayers (**a**) ZMO250; (**b**) ZMO350; (**c**) ZMO/CdS250; and (**d**) ZMO/CdS350. The inset shows the XRD patterns of corresponding CdTe QDs layer. The position of characteristic diffraction peak (111) and the preferred orientation factor f for CdTe are shown in these XRD patterns.

**Figure 5 nanomaterials-12-01523-f005:**
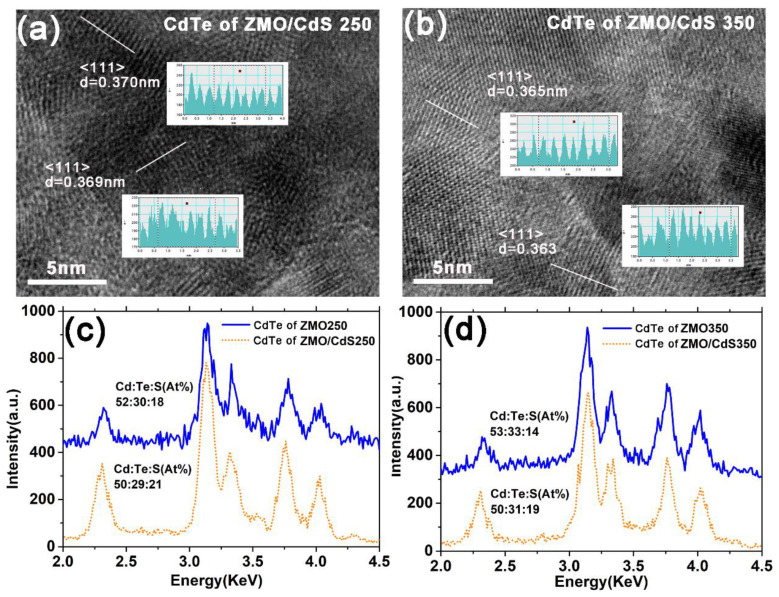
The HRTEM images of (**a**) CdTe in ZMO/CdS250 multilayer and (**b**) ZMO/CdS350 multilayer. The inset of (**a**,**b**) shows the measured interplant distance of CdTe crystalline. EDX patterns of (**c**) CdTe of ZMO/CdS250 multilayer and (**d**) ZMO/CdS350 multilayer. The atomic ratio of Cd:Te:S is listed near EDX patterns.

**Figure 6 nanomaterials-12-01523-f006:**
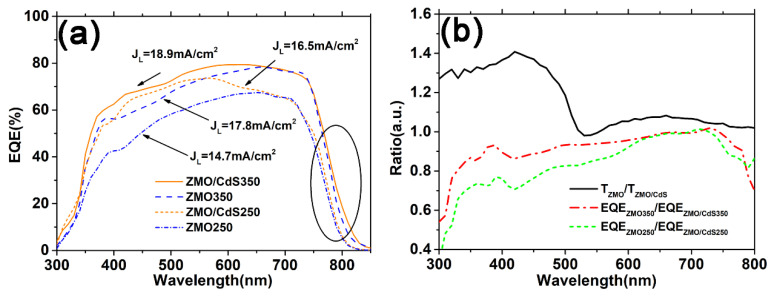
(**a**) The EQE spectrum of the CdTe QDs solar cell fabricated based on the multilayers studied above just adding Au electrode to complete the structure of Glass/ITO/ZMO/CdTe(QDs)/Au and Glass/ITO/ZMO/CdS/CdTe(QDs)/Au. (**b**) The specific value of transmittance of ZMO mono-window layer to ZMO/CdS bi-window layer and the specific value of EQE based on these two different window layers.

**Figure 7 nanomaterials-12-01523-f007:**
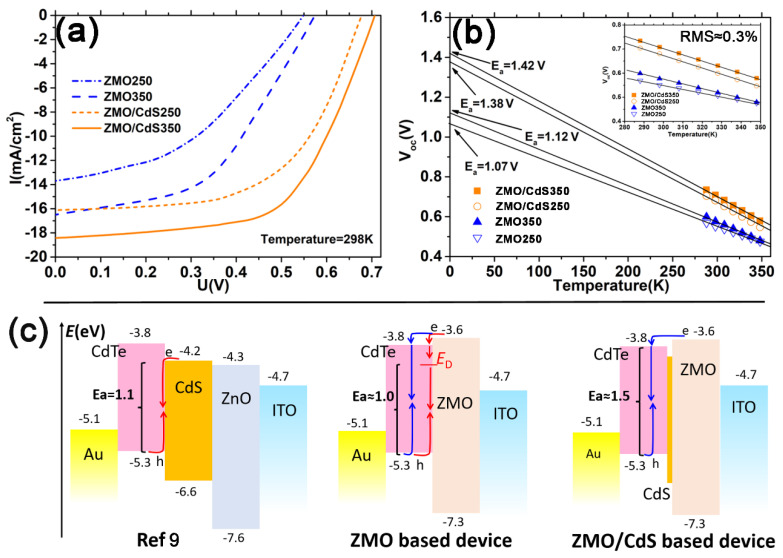
(**a**) Light *J**−V* curves of the corresponding solar cell stated in Figure 6. (**b**) Temperature dependent open circuit voltage *V_oc_* and its linear extrapolation line to 0 K. The inset image is an enlarged view of *V_oc_* vs. *T* between 280 and 355 K. (**c**) Schematic diagram of the diode current density *J*_0_ tuned by band alignment at heterojunction within the CdTe QDs device. The red arrow denotes the interface recombination route and the blue arrow denotes the bulk recombination route. The activation energy *E_a_* corresponding to different recombination mechanisms is shown with bracket.

**Table 1 nanomaterials-12-01523-t001:** (111) Peak position, FWHM, and crystalline size of CdTe QDs film for each multilayer.

CdTe of	ZMO250	ZMO350	ZMO/CdS250	ZMO/CdS350
Position of peak (°)	24.06	24.36	24.06	24.42
FWHM	0.697	0.783	0.680	0.801
Crystalline size(nm)	11.52	10.26	11.81	10.05

**Table 2 nanomaterials-12-01523-t002:** *J-V* parameters of the devices measured in AM1.5 at room temperature 298K.

Device Based On	ZMO250	ZMO350	ZMO/CdS250	ZMO/CdS350
Short circuit current density *J*_s_(mA/cm^2^)	13.70	16.50	16.11	18.42
Open circuit voltage *V_oc_*(V)	0.550	0.577	0.680	0.708
Fill factor *ff*	42.62%	49.31%	59.70%	61.76%
Conversion efficiency *η*	3.21%	4.70%	6.54%	8.06%
Series resistance *R*_s_(Ωcm^2^)	28.0	21.8	19.7	17.5
Shunt resistance *R*_sh_(Ωcm^2^)	280	249	1459	1069

## Data Availability

Data are contained within the article or Appendix A.

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
