# Peer review of "Study on the Aqueous CdTe Quantum Dots Solar Device Deposited by Blade Coating on Magnesium Zinc Oxide Window Layer"

_nanomaterials, 2022, doi:10.3390/nano12091523_

Round 1
Reviewer 1 Report
Very interesting, well written article concerning hot topic of PV. Article presents results of studies on the aqueous CdTe QDs solar cells (SCs) deposited on ZMO by simple method of blade coating. Authors proof an improvement of the SCs efficiency by inserting thin CdS layer between CdTe and ZMO and annealing.
I have only one remark. Authors write in Abstract and Conclusions that it is the spike-like CB alignment between ZMO and CdS which is responsible for this improvement. This issue is not discussed in the article - authors solely refer to the paper [11]. Therefore I recommend to extend discussion on this subject or revise Abstract and Conclusions.
Reviewer 2 Report
The manuscript under title “Study on the aqueous CdTe quantum dots solar device deposited by blade coating on magnesium zinc oxide window layer” by Bin Lv, Xia Liu, Bo Yan, Juan Deng, Fan Gao, Naibo Chen, and Xiaoshan Wu is an original research paper devoted to comprehensive study of the photo-physical properties of the solar cells based on colloidal CdTe quantum dots.
The authors described in detail the original procedure of fabrication of CdTe QD solar cells using blade coating technique and magnesium zinc oxide (ZMO) window layers both ZMO monolayer and ZMO/CdS bilayer. They confirmed experimentally that using of the additional CdS layer one can effectively decrease the interface recombination due to the less lattice mismatch and fast interdiffusion between CdS and CdTe. Then they studied the effect of the annealing on structure and characteristics of the samples. In the main part of the manuscript the authors show a lot of experimental results and make a discussion. The main result of the study is obtaining colloid QD-based solar devices with characteristics comparable with the similar devices developed to date. So, the study indeed provides valuable solutions for low-cost and nontoxic fabrication of aqueous QDs solar cells.
No doubt, the paper is of interest to the research community in the field of optics, photovoltaics, and material sciences and is appropriate for the journal. However, there are few points, which reduce the positive impression of the manuscript.
- The Abstract is overfilled with the technical details and can be shortened.
- The information in the "Results and Discussion" section is difficult to understand without dividing the text into paragraphs: structural, photophysical, photovoltaic measurements, etc. This part of the manuscript also can be improved.
- In the introduction the authors mentioned blade coating technique used for fabrication of QD-based solar cells. This issue concerning fabrication of colloidal QD based materials seems to be very important in other fields of science and technology. I propose the authors to mention some papers in this context:
1) Balazs, Daniel M. et al., Colloidal Quantum Dot Inks for Single-Step-Fabricated Field-Effect Transistors: The Importance of Postdeposition Ligand Removal, ACS Applied Materials and Interfaces, Vol.10, Issue 6, P. 5626 – 5632 (2018). DOI: 10.1021/acsami.7b16882.
2) Zeng, Qunying et al., Efficient larger size white quantum dots light emitting diodes using blade coating at ambient conditions, Organic Electronics, V. 88, 106021 (2021). DOI: 10.1016/j.orgel.2020.106021.
3) Arzhanov A. et al., Incoherent Photon Echo in an Inhomogeneous Ensemble of Semiconductor Colloidal Quantum Dots at Low Temperatures, Bulletin of the Lebedev Physics Institute, Vol. 45, Iss. 3, P. 91 – 94 (2018). DOI: 10.3103/S1068335618030077.
4) Fan, James Z. et al., Micron Thick Colloidal Quantum Dot Solids, Nano Letters, V. 20 (7), 5284 – 5291 (2020). DOI: 10.1021/acs.nanolett.0c01614
- Figure S2. The (a) absorption and (b) photoluminescence spectra of CdTe QDs as functions of reflux time. - There is not any explanation for the observed dependence of the position of the first maximum in the absorption spectrum (in comparison with the behavior of the position of the second maximum) with the increase of reflux time.
- (line 114) Step 5: All multilayers were etched in 0.2% Br/CH3OH solution for 8 s to remove surface oxides and to create a Te-rich surface. - It is not clear how the increase of the content of Te is achieved.
- (line 172) Here the QDs exhibit PL peak at 618nm was chosen as the absorbed materials in following devices. - What is the reason for the choice of the value of 16 hours for the reflux time to growth appropriate QDs? Is it possible to obtain bigger ones with better characteristics using longer growth times?
- J_(L) in Figure 6 needs clarification and it contradicts with I_(L) in line 375.
- Measurement errors are not indicated in Figure 7(b). Large error values combined with a large extrapolation interval can lead to significant differences in the calculated values of Ea.
- For some abbreviations like ITO, MZO, TGA (page 2 and further), EQE (line 126 and further), JPDS (line 132), CPS (figures 1 and 4) AM1.5 (line 366 and Table 2) it is required an explanation for readers who are not experts in the field of study.
- Some chemical notations and abbreviations (subscripts) should be corrected: Br/CH3OH (114), CdCl2 (70, 97, 197, 249) CdTe1-xSx (364), TZMO/TZMO/CdS, EQEZMO/EQEZMO/CdS, EQEZMO250/EQEZMO/CdS250 … (line 321), etc. Please, try to find and check it.
I think that the article can be accepted for publication after minor revision and taking into account the above comments.
Reviewer 3 Report
The manuscript reports on the preparation of magnesium zinc oxide (ZMO) window layer for CdTe quantum dots-based solar cells. Furthermore, ZMO/CdS bi-window layer could decrease the interface recombination effectively. In addition, treatment with CdCl2 and annealing finally resulted in an improvement in the conversion efficiency of the solar cell to 8%. These are predictable results from previously reported bulk solar cells, but it is very important to confirm this experimentally. The paper without doubts will strongly contribute to further progress of the solar cells using CdTe quantum dots and will be of interest for the readership of Nanomaterials. The paper is well written, the experimental material is reliable and clearly discussed, the number and quality of figures is good, the instrumental level is excellent. The Conclusions adequately reflect main findings of the work. Therefore, the reviewer judge that it can be accepted without the need for correction.
